# Estimate of Secondary Porosity from Surface Crossed Square Array Resistivity Measurements

Manuel Joao Matias 

Department of Geosciences, Campus de Santiago, University of Aveiro, 3810-193 Aveiro, Portugal; mmatias@ua.pt

**Abstract:** The secondary porosity of rocks and formations plays an important role in water exploration and exploitation in crystalline rocks. Furthermore, cracks and voids are paths for contaminated fluid propagation and, thus, their location can be very important in environmental studies. Usually, secondary porosity is estimated from well logging observations, but some previous works have pointed out that azimuthal resistivity measurements and the square array can be used to estimate this parameter. Herein, the use of the square array and of the crossed square array will be investigated to estimate the secondary porosity. Thus, the analogue model and field resistivity data obtained with those arrays are discussed and interpreted. The analogue model simulates contacts between the isotropic and anisotropic media and between two different anisotropic media. The field data are from an area where steeply dipping formations prevail and contacts between the isotropic and anisotropic media are present. The calculated secondary porosity values are compared with the lateral changes in the resistivity and anisotropy, and pseudo sections are shown to aid the interpretation. In addition, 2D resistivity models for the field survey are produced to provide a better image and interpretation of the data. The overall results and limitations are discussed, and data acquisition procedures are proposed to optimize the field work.

**Keywords:** secondary porosity; resistivity; surface; measurements; crossed square; array; groundwater



## 1. Introduction

The porosity and permeability of rocks are the main properties that allow fluid propagation in the earth's interior. The porosity of rocks [1] during their original formation is called primary porosity and is affected by the shape, packing and distribution of grains. Rock cementation can affect the primary porosity and is particularly important in sedimentary rocks. However, most rocks have little or no primary porosity when they are formed. As a result of fracturing and weathering, voids will be produced after rock formation. This additional porosity acquired after rock formation is called secondary porosity [1]. The secondary porosity increases the rock porosity and, together with the primary porosity, will produce the so-called effective porosity of a rock. Thus, the secondary porosity plays an important role in fluid propagation in the earth's interior, particularly in formations that have no primary porosity.

Anthropic activities, such as caving and mining, will also produce cavities that will contribute to fluid propagation.

Therefore, the estimate of the secondary porosity can contribute to the location of fractures, cracks, voids and cavities (man-made or from natural chemical dissolution processes) and, hence, to the location of fluid propagation paths in the earth's interior [2], with obvious applications in water exploration and exploitation in crystalline environments, as well as in the investigation of contaminants propagation.

Secondary porosity is usually estimated from well logging studies [3,4], but it has been proposed to estimate it from surface resistivity measurements [5,6]. Most of these works use the azimuthal resistivity array technique, in which one array, usually linear, is rotated around a central point [7–11]. The square array has also been used to perform

the azimuthal resistivity technique [12–14], rotating it around the center, similarly to the traditional azimuthal technique for linear arrays.

These techniques, both for linear and square arrays, require space and are time consuming if the complete range of orientations are investigated.

The square array has extreme orientation responses for orientations of 0° and 45° in relation to the geological strike [15], whilst the crossed square array has proven to be orientationally stable and to provide accurate estimates for the apparent anisotropy, effective vertical anisotropy "*n*" and the strike of concealed structures [16,17].

Therefore, this work aims to investigate the use of square and crossed square arrays in the estimate of the secondary porosity from surface resistivity measurements. As previous works [6,8,12] discuss the data from geological settings poorly known, to prove the suitability of the method, resistivity measurements from analogue models and known geological settings are discussed in order to investigate and assign secondary porosity values to known structures.

Therefore, the data obtained over an analogue model, representing the contact between one isotropic and an anisotropic medium and a further contact between two anisotropic media, will be discussed.

Subsequently, two field resistivity traverses, obtained with the use of the crossed square array, will be presented and the behavior of the computed values for the secondary porosity will be evaluated against the computed effective vertical anisotropy values and the known geological structure of the sites.

The data will be discussed in the form of 2D pseudo sections, models and profiles for a better understanding of the benefits of the technique.

Finally, field survey guidelines will be given to optimize the field work and data collection.

## 2. Crossed Square Array of Electrodes and Secondary Porosity Estimates

The crossed square array [15] is composed of two square arrays rotated clockwise through 45°, Figure 1.

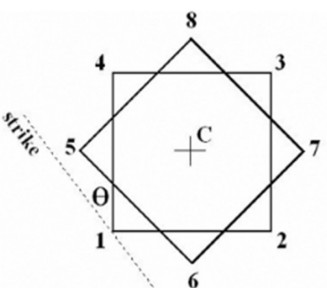

| CONFIGURATION | ELECTRODE POSITIONS | | | | | | | | OBSERVED RESISTANCES |
|---|---|---|---|---|---|---|---|---|---|
| | 1 | 2 | 3 | 4 | 5 | 6 | 7 | 8 | |
| $\alpha$ | A | M | N | B | | | | | $R_\alpha = R_1$ |
| $\beta$ | A | B | N | M | | | | | $R_\beta = R_3$ |
| $\gamma$ | A | M | B | N | | | | | $R_\gamma$ |
| $\alpha'$ | | | | | A | M | N | B | $R_{\alpha'} = R_2$ |
| $\beta'$ | | | | | A | B | N | M | $R_{\beta'} = R_4$ |
| $\gamma'$ | | | | | A | M | B | N | $R_{\gamma'}$ |

**Figure 1.** The crossed square array of electrodes and measurements.

The first square, electrodes 1 to 4 in Figure 1, provides three readings, the $\alpha$, $\beta$ and $\gamma$ resistances for square array 1. The second square array, 5 to 8 in Figure 1, also provides another three readings, the $\alpha$, $\beta$ and $\gamma$ resistances, but for square array 2. As square array 1 and square array 2 are rotated 45°, if the array orientation, $\theta$ in Figure 1, is in accordance with the geological strike, that is $\theta = 0°$, square array 1 and square array 2 will provide the results for the extreme orientations of the array [15]. In this case, owing to the paradox of anisotropy [15,16], the computed resistivity values for square array 1 and square array 2 will correspond to the maximum and minimum of the resistivity measurements in the region.

Thus, in these circumstances, for square array 1 (1, 2, 3 and 4 in Figure 1), the resistivity is given by [18]:

$$\rho_{1234} = \rho_{max} = \frac{2\pi a}{2 - \sqrt{2}} \times \frac{R_1 + R_3}{2} \tag{1}$$

For square array 2 (5, 6, 7 and 8 in Figure 1), the resistivity is given by [18]:

$$\rho_{5678} = \rho_{min} = \frac{2\pi a}{2 - \sqrt{2}} \times \frac{R_2 + R_4}{2} \tag{2}$$

where "a" is the square side. In SI, the resistivity unities are ohm $\times$ m.

The crossed square array is particularly suited to carry out surveys over inhomogeneous and anisotropic terrains. Over a uniform anisotropic half space, in Figure 2, it is defined [15] as a mean resistivity "$\rho_m$" and the non-dimensional anisotropy coefficient "$\lambda$", so that:

$$\rho_m = \sqrt{\rho_l \, \rho_t} \tag{3}$$

and

$$\lambda = \sqrt{\rho_l \, / \rho_t} \tag{4}$$

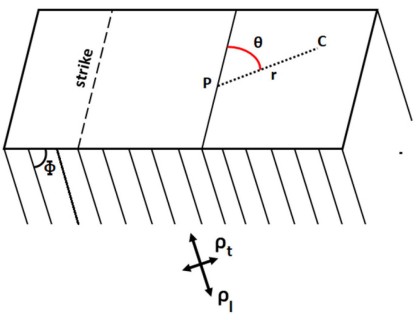

**Figure 2.** Uniform anisotropic half space.

Over the model in Figure 2, the electrical potential "V" at a point "P" distant "r" from a current source "C" is given by [15]:

$$V = \frac{I\rho_m}{2\pi r} \times \left(1 + \left(\lambda^2 - 1\right) sin^2\theta sin^2\phi\right)^{-1/2} \tag{5}$$

In Equation (5), the anisotropy "$\lambda$" and the dip "$\phi$" cannot be separated. Therefore, they were combined [15] in a single non-dimensional parameter "$n$", the so-called effective vertical anisotropy:

$$n = \left(1 + \left(\lambda^2 - 1\right) sin^2\phi\right)^{-1/2} \tag{6}$$

The effective vertical anisotropy varies between 1 and "$\lambda$" and it assumes the value of "$\lambda$" for the vertical dipping layers.

As the crossed square array provides measurements in four orientations, rotated through 45°, the processing of its data enables us to obtain estimates for the apparent resistivity, the effective vertical anisotropy "$n$" and strike, θ, can be found in [19].

Finally, the secondary porosity estimates, $\phi$, can be calculated from [5,6,12]:

$$\phi = \frac{3.14 \times 10^4 \, (n-1)\left(n^2 - 1\right)}{n^2 C(\rho_{max} - \rho_{min})} \tag{7}$$

where $\phi$ is the fraction of the volume of voids and fractures in the total volume of the formations (if this fraction is to be presented in percentage it must be multiplied by 100), C is the conductivity of groundwater (microsiemens/cm) [5], "$n$" is the non-dimensional effective vertical anisotropy as computed from the crossed square array and $\rho_{max}$ and $\rho_{min}$ are the maximum and minimum apparent resistivity values in the area under investigation.

Therefore, if the crossed square array is oriented in accordance with the geological strike, the apparent resistivities calculated from square array 1 and square array 2 will be the maximum and minimum resistivities of the area and can be used in Equation (7) to compute the secondary porosity, as previously stated. However, the application of

Equation (7) will provide a rough estimate of the secondary porosity as the presence of clay in the fractures can lead to incorrect results [5].

A closer look at the secondary porosity in Equation (7) raises some problems. In fact, over homogeneous or near homogeneous ground, the values of $\rho_{max}$ and $\rho_{min}$ will be very close and, for homogeneous ground, they should be equal. On the other hand, in these conditions, the "*n*" values should approach the value 1. Thus, it is possible that anomalous large values greater than 1 for the calculated secondary porosity can be produced and, in these cases, it is necessary to check the values to identify these situations, as the calculated values have no geological meaning.

Furthermore, it has been reported [15,17] that, in some cases, anisotropy axes can rotate and produce the so-called oblate anisotropy; that is, strike values at 90° to the true strike. In these cases, $\rho_{max}$ becomes $\rho_{min}$ and $\rho_{min}$ becomes $\rho_{max}$; then, negative secondary porosity values will be computed. This occurs at low "*n*" values [15,17] and, in these cases, the values must also be checked.

Another point for discussion in Equation (7) is the constant C. This equation has been proposed by several authors [5,6,12], and for some [5], C is the groundwater conductivity, whilst for others [6,12], C is the groundwater specific conductance, both in microsiemens/cm. This is a supplementary problem as either value should be measured on site from groundwater samples, which are not always available. Furthermore, it is possible that the C varies in a region as a result of contamination. Therefore, values are often taken from tables or any other available information [20,21]. Herein, electrical conductivity values were used for C and, in the model case, this value was calculated from the brine itself; but, for the field survey, the C values were taken from the general information in the area.

As the final secondary porosity values depend on C, this is another setback that limits the estimates. However, if C is constant, it will affect all the values of a survey in the same manner; therefore, it will always be possible to look for the lateral changes, variations and local maximum values of the computed secondary porosity values instead of absolute values. If this is the case, the secondary porosity values will be very useful for identifying the features of particular interest (fractures, voids, schistosity) that can be associated with paths for underground fluid circulation.

Now it is proposed to carry out this analysis over an analogue model and in a field survey to investigate the problems described above.

## 3. Secondary Porosity Computed over an Analogue Model

Crossed square array measurements were carried out over a model, Figure 3, similar to those used in [22].

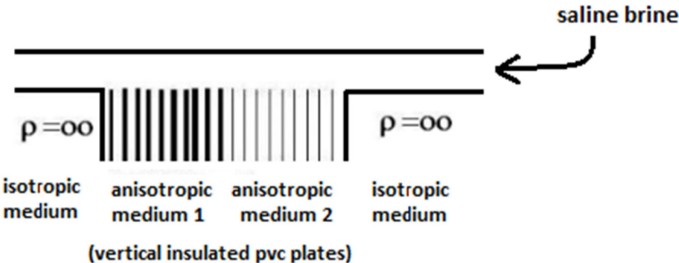

**Figure 3.** Analogue Model.

This model was built by adding a polyvinyl chloride (PVC) insulated block, 1 m wide, 1 m long and 20 cm high, on the left, to a sequence of vertical PVC plates, 6 mm thick, 1 m wide and 20 cm high, on the center-left, and to a sequence of vertical PVC plates, also 6 mm thick, 1 m wide and 20 cm high, coated with aluminum foil, on the center-right. The space between the plates, 6 mm thick laminae, is filled with saline brine. Finally, another isotropic insulating block was added at the far-right, with similar dimensions to the block on the left. The block on the left acts as homogeneous ground, and the other blocks, to the right, act as

anisotropic blocks of different anisotropies. The block on the far-right acts as an isotropic block, homogeneous ground, similarly to the one on the far-left. The entire model was submerged in saline brine, with a conductivity of 2836 microsiemens/cm, at a depth of 3 cm.

This model intends to simulate complex concealed geological settings, which are difficult to address in numerical models. The contacts between the isotropic blocks and the anisotropic blocks intend to emulate the geological contacts or faults separating isotropic formations, such as non-layered or semi horizontal media, from steeply dipping formations, such as schists, with higher secondary porosity properties. Therefore, if the secondary porosity can be measured. it should be caused by the contact itself and by the properties of the anisotropic medium (schists). On the other hand, the contact between the two different anisotropic media simulates the contact or fault between two different steeply dipping formations, such as two different types of schists. In this case, the secondary porosity will be caused by the contact itself and by the properties of the two media. As the two media are different, the hope is to obtain a different answer for the contact and for the two media.

Crossed square array measurements were conducted over this model using square array sides ranging between 1.41 cm and 45.2 cm with an expansion rate of $\sqrt{2}$. The use of this expansion rate enables six field or model resistivity measurements for the first decade, seven for the second and six for the third decade of measurements. Therefore, very good and uniform data spatial sampling is obtained. Furthermore, as for the sounding interpretation, these spacings are usually plotted on a logarithmic scale, the expansion rate of which provides a uniform distance between the spacings used and, hence, a much better overall sounding curve definition. The crossed square array was oriented in accordance with the orientation of the plates, which is $\theta = 0°$. The soundings were carried out every 4 cm; however, in the vicinity of the contacts, this distance was reduced to 2 cm.

The relation between the calculated strike and the effective vertical anisotropy "$n$" is shown on the left of Figure 4.

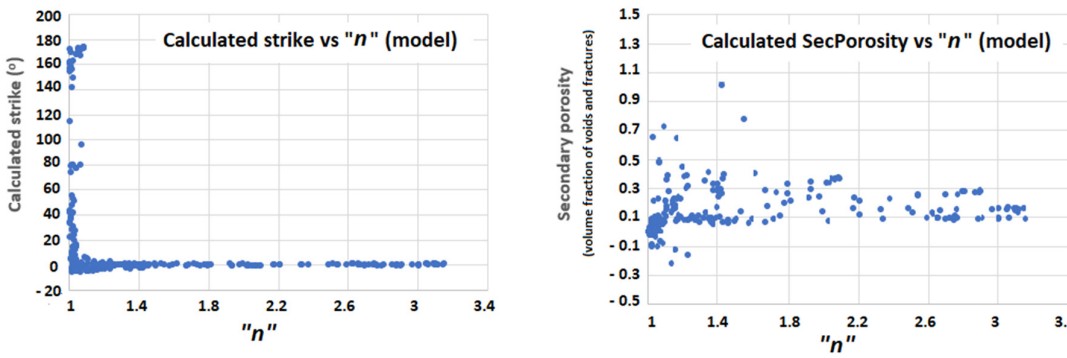

**Figure 4.** Relation between the calculated strike and "$n$" (on the left) and relation between "$n$" and the calculated secondary porosity (on the right).

As can be seen on the left of Figure 4, very good strike values are obtained for a wide region of "$n$" values, in agreement with previous studies [16,17,19]. Thus, $\theta = 0°$ and square array 1 will provide the values for $\rho_{max}$ and square array 2 will provide the values for $\rho_{min}$; therefore, the secondary porosity Equation (1) can be applied.

On the right of Figure 4, for "$n$" values greater than 1.2, the calculated secondary porosity remains stable and shows one level with values varying between 0.1 and 0.5 for "$n$" values until 1.9. Another level for values ranging between 0.1 and 0.3 is shown for "$n$" values greater than 2.2.

As expected for lower "$n$" values, the computed secondary values vary largely, and negative values are depicted. These values were checked, and they correspond to the smallest spacings, where the errors are larger. No cases of oblate anisotropy were found; that is, strike determinations at 90° to the model strike. In all cases, the difference between $\rho_{max}$ and $\rho_{min}$ is very small and these negative secondary porosity values must be seen

as noise. Therefore, at low "$n$" values, estimates for the secondary porosity are difficult and meaningless.

To further this analysis, the profiles for the square array sides of 4 cm, 11 cm and 32 cm are shown in Figure 5.

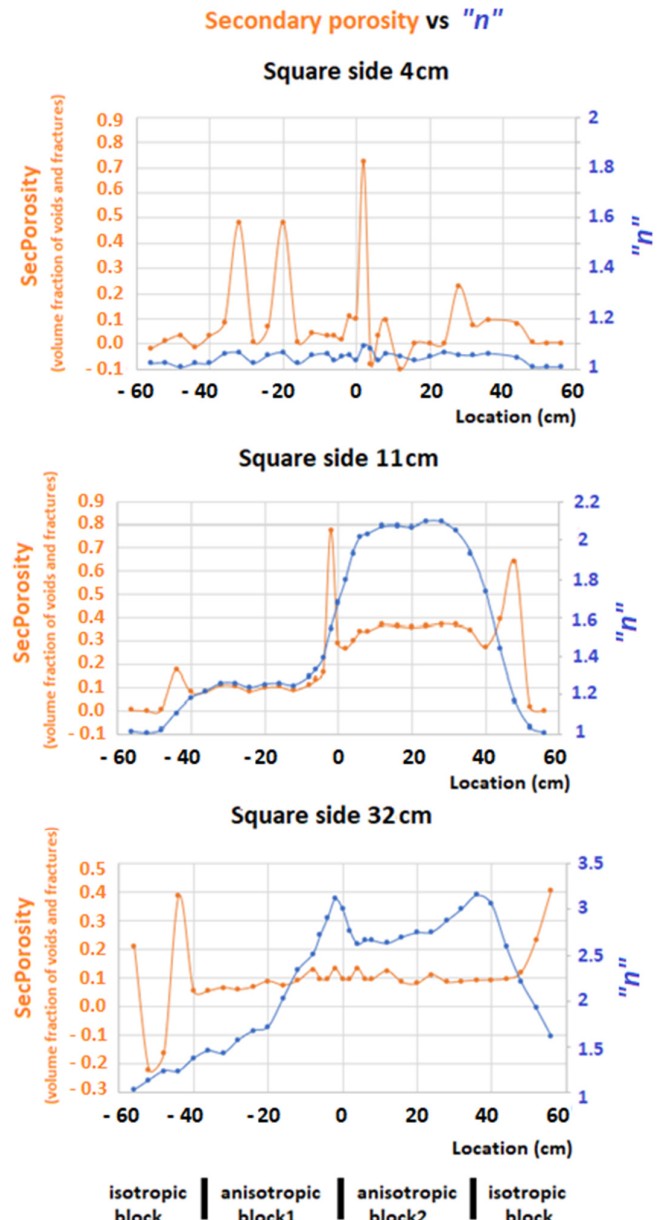

**Figure 5.** Secondary porosity vs. "$n$": square side 4 cm (top); square side 11 cm (center); square side 32 cm (bottom). (YY' axis: fraction of the total volume occupied by voids and fractures).

Below the profiles, the lowest part of Figure 5 also shows the limits of the different blocks; that is, the boundaries between the two isotropic blocks, left and right, and the anisotropic blocks at locations −38 cm and +38 cm, as well as the boundary between the two anisotropic blocks at location 0 cm.

The top graphs refer to a square side of 4 cm and, at this spacing, it is expected that most of the signals arise from the top brine. In fact, the "$n$" values are lower than 1.1 and the secondary porosity estimates vary largely; therefore, the results are meaningless.

The graph at the center of Figure 5 refers to a square side of 11 cm. In this case, the computed values for "$n$", depicted in blue, and the secondary porosity, in orange, reveal

the boundaries of the four blocks clearly. The "*n*" curve reveals the nature of the blocks; that is, values close to 1 over the isotropic blocks and two anisotropic blocks with different anisotropy. The anisotropic block to the right, with the PVC coated with aluminum foil, shows higher anisotropy values.

The secondary porosity values clearly show the positions of the contacts between the blocks. However, the contacts between the isotropic and the anisotropic blocks seem to be slightly shifted towards the isotropic block, as the positions for the maxima are at points −42 cm and +44 cm. The center maximum, at position −1 cm, gives a more accurate position of the contact between the two anisotropic blocks.

Another characteristic of the secondary porosity curve is that it provides stable values around 0.1 over anisotropic block 1, whilst the values are close to 0.4 over anisotropic block 2 and, therefore, it differentiates the two blocks.

Considering that the model is composed of an alternating sequence of 6 mm thick vertical insulating PVC plates and 6 mm thick vertical brine laminae, the volume of the "voids" is 50% of the total volume as the aluminum foils are very thin and should not alter the thickness of the brine laminae. Therefore, it can be argued that the secondary porosity over the anisotropic blocks should be 0.5. This value is not far from the values calculated for anisotropic block 2. However, for anisotropic block 1, at the bottom of Figure 3, the calculated secondary porosity values, although very stable, are around 0.1, which are far from the expected values. Therefore, if the brine laminae were considered voids, the computed secondary porosity values do not reach the expected values. Nevertheless, they are very stable and consistent over the two anisotropic blocks, separating them as well as their boundaries.

Finally, the bottom graphs in Figure 5 show the results for a square side of 32 cm. The "*n*" curve, in blue, depicts a maximum at point 0, which is the boundary between the two anisotropic blocks, and another peak near position 40 cm, which is the contact between anisotropic block 2 and the isotropic block on the right part of the model. However, the secondary porosity curve, in orange, depicts a central part with stable values around 0.1 that corresponds to the two anisotropic blocks. The ends of the anisotropic blocks are also marked near positions −38 cm and 42 cm. Thus, the secondary porosity curve for this array size seems to be too big to discriminate the difference but is able to detect the limits.

With all the computed secondary porosity values, a pseudo section was constructed, as shown in Figure 6. Below the pseudo section, the black lines mark the limits of the blocks to help the discussion.

The data in Figure 6 clearly show the boundary between the isotropic block and the anisotropic block 1, on the left-hand side of the model. The second anisotropic block 2 is also clear as it provides higher secondary porosity values than the anisotropic block 1 on its left. The boundary between these two blocks is depicted for spacing values less than 20 cm. To the right of the model, the contact between the anisotropic block 2 and the isotropic block can only be seen at larger spacings, at position 40. For shorter spacings, the computed secondary porosity values are very high. These high values for spacings less than 20 and location larger than 45 are due to very close values for $\rho_{max}$ and $\rho_{min}$ as they are recorded over the isotropic block, and thus have no meaning.

In conclusion, the absolute secondary porosity values are lower than expected, but respond clearly to the limits of the blocks and can be related to the nature of the model, that is, they differentiate the blocks of different anisotropies.

Therefore, these results reveal the importance of the lateral changes in the secondary porosity values rather than its absolute values. However, optimal information is obtained from certain spacings only, in this case 11 cm.

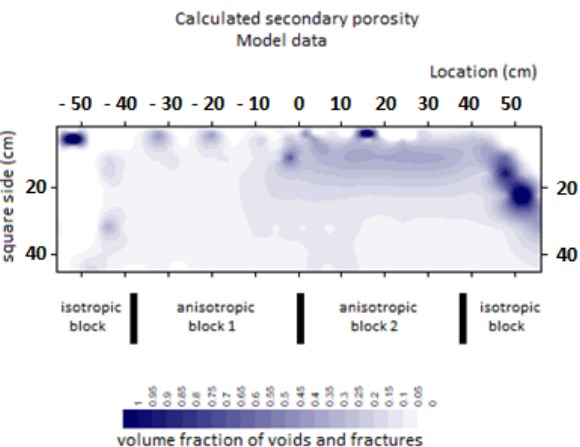

**Figure 6.** Pseudo section of secondary porosity values. (Secondary porosity as the fraction of the total volume occupied by fractures and voids).

## 4. Field Survey

The data from a field survey [16] are now reprocessed to extract the information on the secondary porosity. Furthermore, as these data were interpreted in the 1D mode [16], now they will be modelled in the 2D mode, using RES2DINV [23,24], to assist the secondary porosity interpretation.

Field work was carried out in the Chapel le Dale Valley, near Ingleton, Figure 7, latitude 54°9′ N, longitude 2°28′ W, northwest England. The location of the two profiles, CLD1 and CLD2, is depicted in Figure 7, and the geology of the area is well-known [25,26].

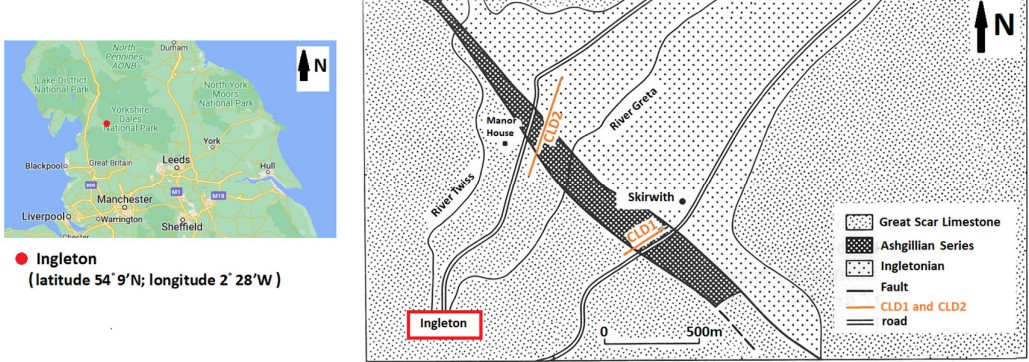

**Figure 7.** Survey area location and geology.

Chapel le Dale Valley is a glacial valley where Carboniferous Limestone was weathered away, so that older basement formations are exposed or covered by drift of variable thicknesses. Therefore, the area in Figure 7 is characterized by Limestones, with a near horizontal layering; steeply dipping formations, the Ashgillian and Ingletonian series; vertical and near vertical contacts; dykes covered by drift of varying thickness, and with a general northwest-southeast strike.

Therefore, if the crossed square array surveys are set with electrode 1 in Figure 1, oriented towards the North, the orientation, θ in Figure 1, of the array in relation to the geological strike will be 0°. In this case, square array 1 will provide the values for $\rho_{max}$, whilst square array 2 will give the values for $\rho_{min}$, and Equation (1) can be used.

The Chapel le Dale 1 (CLD1) traverse, shown in Figure 7, consists of 13 crossed square array soundings up to a spacing of 45.2 m, with the traditional square array expansion factor of $\sqrt{2}$. Chapel le Dale 2 (CLD2) used 30 crossed square array soundings with spacings up to 64 m with the same expansion factor as that for CLD1. In all cases, the crossed square array was oriented in accordance with the local geological strike, which is θ = 0°, and secondary

porosity calculations were carried out with an electrical conductivity value of 250 microsiemens/cm. The distance between adjacent soundings is not constant, as it depends on local obstacles but, nevertheless, a thorough coverage of the area was accomplished.

### 4.1. Chapel le Dale 1 (CLD1)

The location of CDL1 traverse is shown in the lower part of Figure 7.

The relation between the calculated strike and "$n$", as well as the relation between "$n$" and the calculated secondary porosity for the CLD1 data, are depicted in Figure 8.

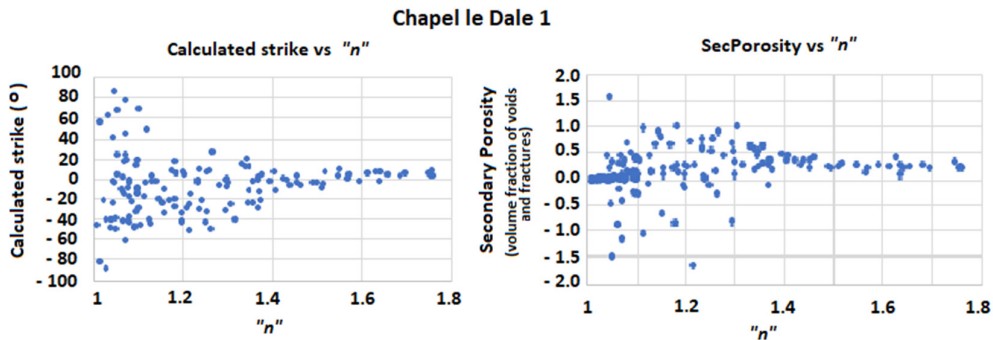

**Figure 8.** Relation between the calculated strike and "$n$", on the left, and relation between "$n$" and the calculated secondary porosity, on the right, for CLD1 data.

As it can be seen, the graph on the left of Figure 8 shows that as the "$n$" values increase, the computed strike tends to $0°$, confirming that the crossed square array is well oriented in accordance with the geological strike. Thus, square array 1 provides the values for $\rho_{max}$ and square array 2 provides the values for $\rho_{min}$, as mentioned before.

On the right part of Figure 8, for higher "$n$" values, the computed secondary porosity values vary between 0.2 and 0.5. For smaller "$n$" values, that is "$n$" < 1.2, the secondary porosity values do not show any tendency, and negative values are recorded. These values were checked and, similarly to the model case, no oblate anisotropy was found. The negative values are all recorded at the smallest spacings, where placement errors and ground inhomogeneities in the vicinity of the electrodes usually introduce noise in field readings. Therefore, these values have no geological meaning.

To carry out field data inversion, the square array spacings were converted into equivalent Schlumberger spacings using the equivalence parameters in [12,18], and the apparent resistivity values obtained from the crossed square array [19]. The data were inputted in the form of the general array, an inversion was carried out and the results are depicted in Figure 9.

The rms error of the model is less than 1% and, as depicted in the bottom block section, it is easy to correspond the modelled resistivity blocks, bottom of Figure 9, to the known geology, Figure 7.

Thus, the left end of the model corresponds to the resistive Great Scar Limestone. The contact between these formations and the more conductive Ashgillian series is shown at position 40 m, and between positions 70 and 80 m, an insulating dyke is also interpreted. To the right of the dyke, for positions greater than 80 m, the Ingletonian formations, covered by a thicker drift, are interpreted.

The computed secondary porosity for square array spacings 4, 22.6 and 45.2 m is depicted in Figure 10, together with the corresponding "$n$" values.

The graph on top, for a 4 m spacing, shows secondary porosity values varying over the positions of the Great Scar Limestone, that is, for locations less than 40 m. In this region, the "$n$" values are also low (less than 1.1), revealing the homogeneous nature of the ground or the near horizontal layering of the Great Scar Limestone.

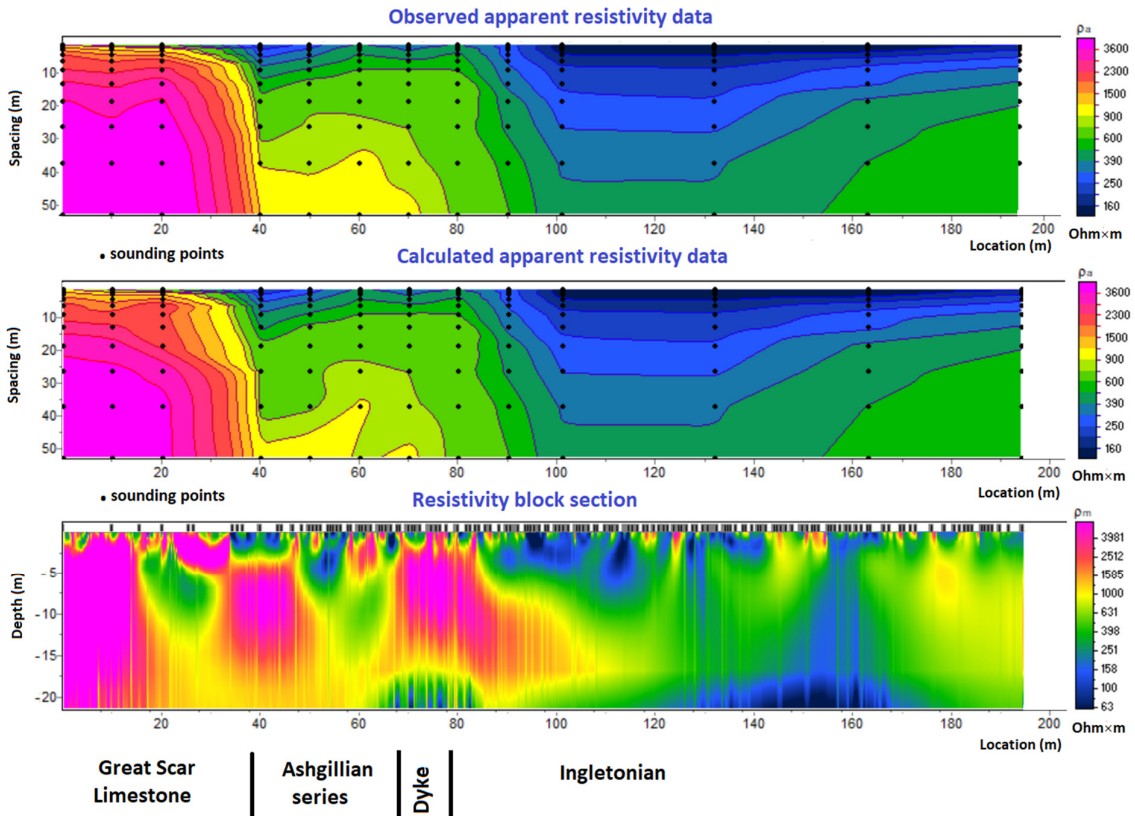

**Figure 9.** CLD1 inverted data.

The dyke position between 70 and 80 m marks the increase in the secondary porosity values. Furthermore, a peak of "*n*" values is clearly developed at these positions. To the right of the dyke position, locations greater than 80 m show very stable "*n*" values and around 1.1, whilst the secondary porosity values are decreasing. These values reach a maximum over the dyke and, as the drift thickness increases, as shown in Figure 9, to the right of the dyke, the secondary porosity decreases at these shorter spacings.

The central graphs show values for a square array spacing of 22.6 m. Again, the positions corresponding to the Great Scar Limestone, less than 40 m, depict low values both for the secondary porosity and "*n*".

The dyke position marks an increase in both values; the "*n*" values reach a peak and the secondary porosity shows values between 0.2 and 0.4. Thus, the secondary porosity is responding to the nature of the ground, that is, to the schistosity of the Ingletonian formations.

Finally, the bottom graph, for a square spacing of 45.2 m, shows that the contact between the Great Scar Limestone and the Ashgillian series is marked by a sharp increase in the secondary porosity. Furthermore, the secondary porosity values are stable, around 0.25, from the position of the dyke to the right.

Finally, the behavior of "*n*" follows the geology of the area.

Therefore, in the case of CLD1, the secondary porosity values reveal the contact between the Great Scar Limestone, at the largest spacings, but fail to discriminate between the dyke and the Ingletonian formations to its right. However, the "*n*" values are proven to be a valuable help in this discrimination.

The pseudo section for the calculated secondary porosity values is shown in Figure 11.

This pseudo section provides more detailed information than that obtained from the profiles in Figure 10. Thus, the three limits between the formations are shown and correlate well with their position depicted below the pseudo section in Figure 11. Unusually high secondary porosity values are shown near the surface and, after inspection, correspond to the very close values for the calculated $\rho_{max}$ and $\rho_{min}$.

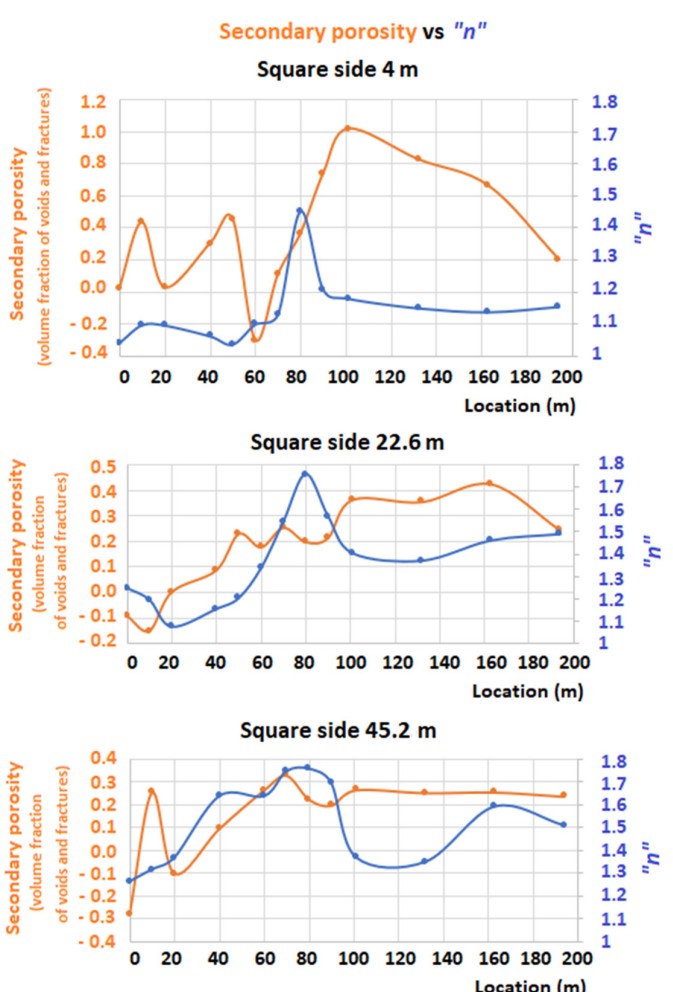

**Figure 10.** Secondary porosity vs. "*n*" for CLD1: square side 4 m (top); square side 22.6 m (center); square side 45.2 m (bottom). (YY' axis: fraction of the total volume occupied by voids and fractures).

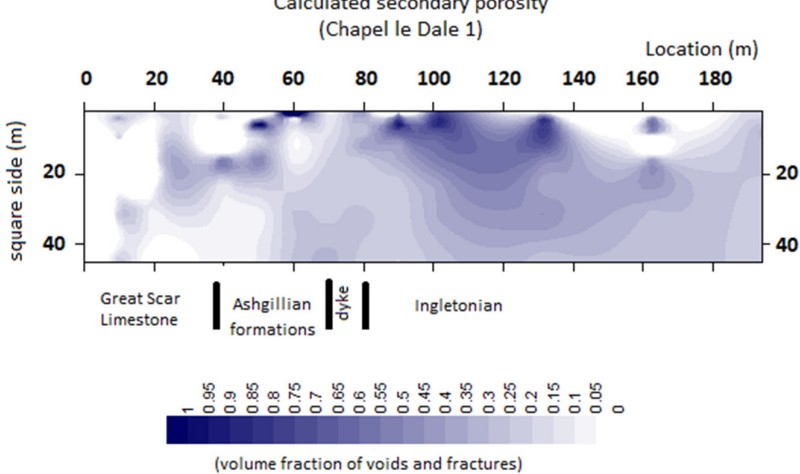

**Figure 11.** Pseudo section for the secondary porosity in CLD1. (Secondary porosity as the fraction of the total volume occupied by fractures and voids).

### 4.2. Chapel le Dale 2 (CLD2)

The location of the CDL2 traverse is shown in the upper part of Figure 7.

The relation between the calculated strike and "*n*", and the relation between "*n*" and the calculated secondary porosity for the CLD2 data, are depicted in Figure 12.

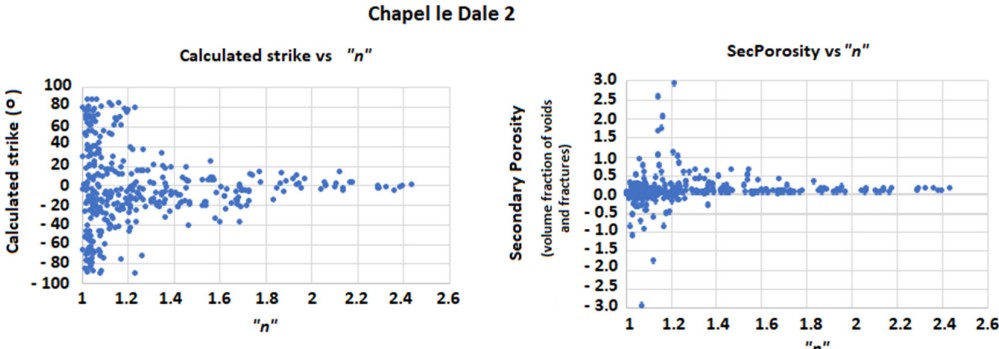

**Figure 12.** Relation between the calculated strike and "*n*", on the left, and relation between "*n*" and the calculated secondary porosity, on the right, for CLD2 data.

As depicted in Figure 12, the calculated strike for the geological formations tends to θ = 0° as the "*n*" increases. Therefore, as in the case of CLD1, the resistivities calculated with square array 1 and square array 2 can be used to calculate the secondary porosity values.

The secondary porosity values are depicted on the right-hand graph of Figure 12. As can be seen, there is a region of values between 0.1 and 0.5 for the "*n*" values varying between 1.3 and 1.6. There is another region of secondary porosity values around 0.2 for "*n*" values larger than 1.8. Unusually high secondary porosity values, positive and negative, are again calculated for the lower "*n*" values. As with the previous case, they correspond to the smallest spacings and should be attributed to very small resistivity variations or the local inhomogeneity of the ground near the electrodes, as no oblate anisotropy values were found after checking the data. Hence, these values have no geological meaning, as in the previous case.

The inversion of the CLD2 data was carried out in the same manner as CLD1. The inverted results are depicted in Figure 13.

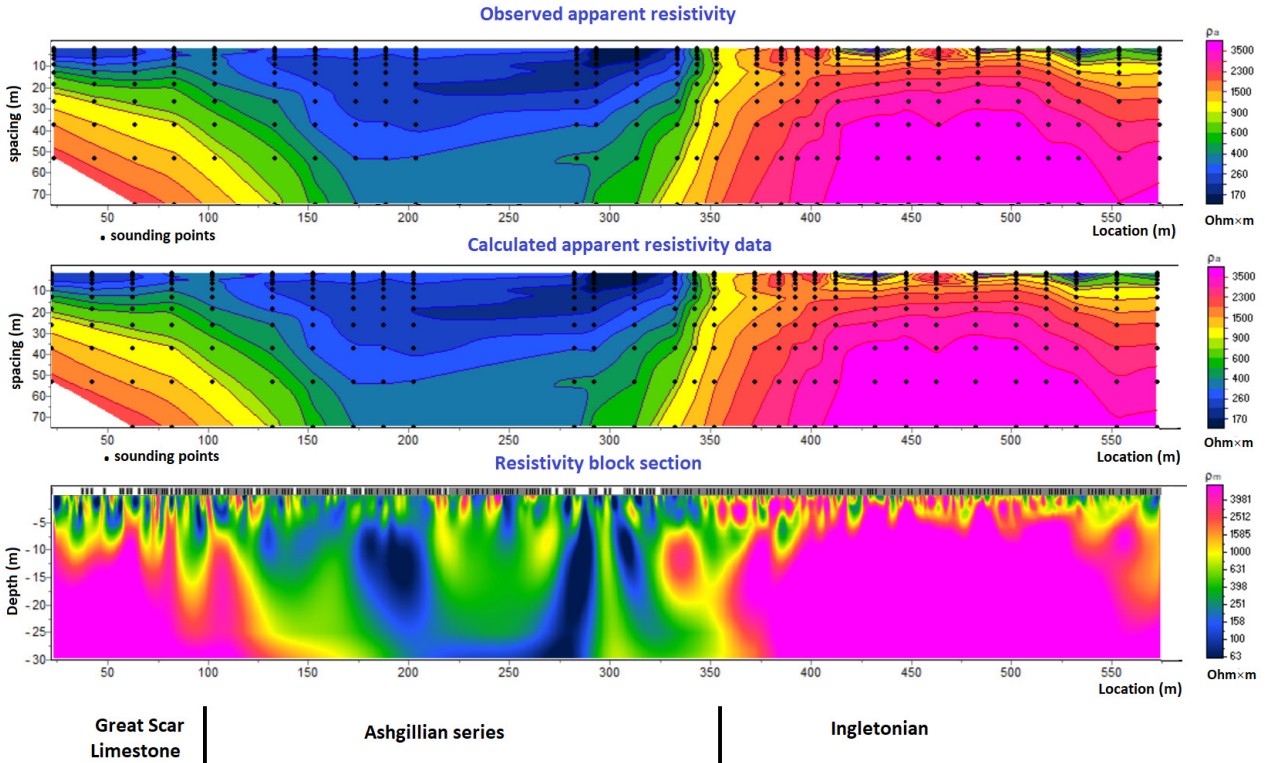

**Figure 13.** CLD2 inverted data.

The rms error for the inverted data is 0.5% and, as depicted in the bottom block section, it is easy to correspond the modelled resistivity blocks to the known geology. That is, the left-hand side of the resistivity blocks in the bottom section correspond to the Great Scar Limestone until location 100. The Ashgillian formations are present from location 100 m to location 360 m and, to the right of location 360 m, the Ingletonian formations are found. From the model, the Ashgillian formations are covered by a thicker drift.

The computed secondary porosity for square array spacings of 4, 32 and 45.2 m is depicted in Figure 14, together with the corresponding "*n*" values.

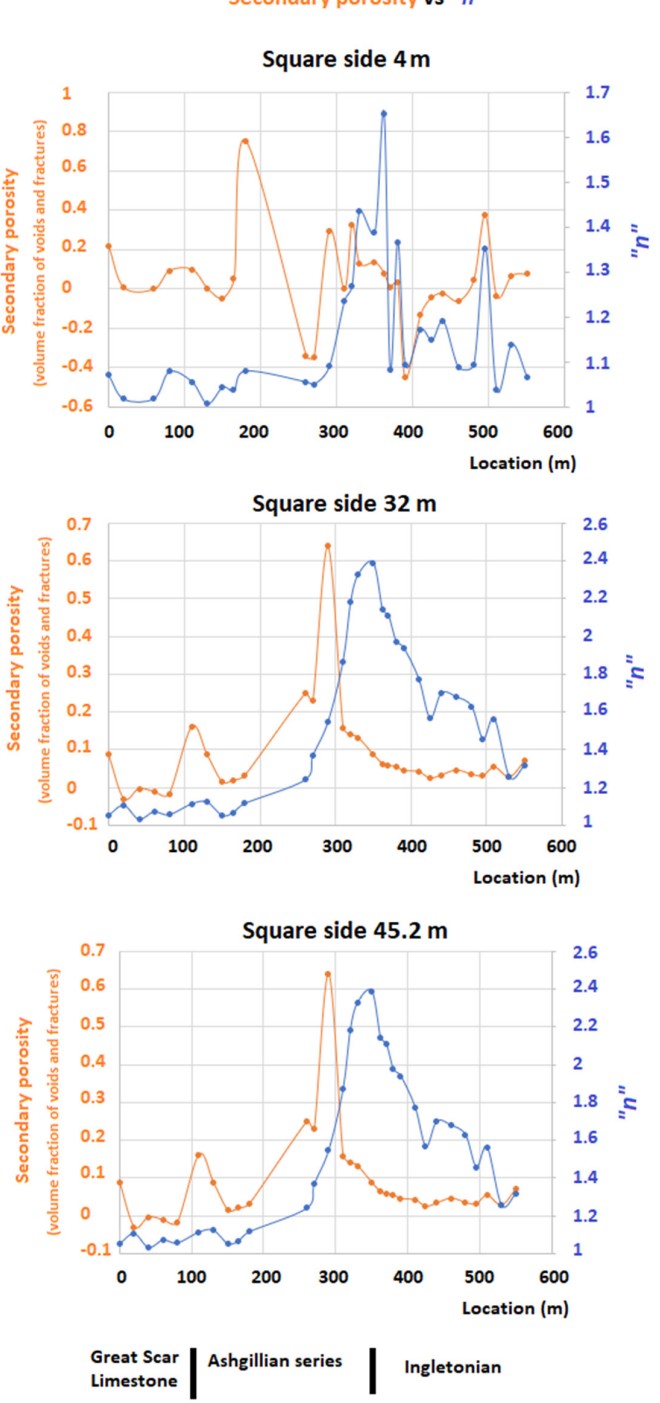

**Figure 14.** Secondary porosity vs. "*n*" for CLD2: square side 4 m (top); square side 32 m (center); square side 45.2 m (bottom). (YY' axis: fraction of the total volume occupied by voids and fractures).

The top graph, square side 4 m, depicts varying values for the secondary porosity, as in the previous cases. The "$n$" values provide maximum values near the contact between the Ashgillian and the Ingletonian formations and, apart from that, little can be said about their behavior.

The central graph shows the data for a square side of 32 m. As can be seen, the secondary porosity values reveal the contact between the Great Scar Limestone and the Ashgillian series at position 100 m.

Further to the right, the contact between the Ashgillian series and the Ingletonian formations seems to be shifted to position 300 m. The region corresponding to the Ashgillian formations shows secondary porosity values varying between 0.05 and 0.2. On the other hand, the region corresponding to the Ingletonian formations shows secondary porosity values in the region of 0.05.

The lowest graph depicts the data for a square side of 45.2 m. In this case, the secondary porosity shows the contact between the Great Scar Limestone and the Ashgillian series at position 100 m, whilst the contact between the Ashgillian and the Ingletonian is shifted towards position 300, as in the case of the central graph (square side 32 m). The secondary porosity values are in the same range as those for a square side of 32 m (central graph).

The "$n$" values graph reveals the contact between the Ashgillian and the Ingletonian at position 360 m, but is not clear for the contact between the Great Scar Limestone and the Ashgillian.

Finally, Figure 15 shows the pseudo section for the secondary porosity values in CLD2.

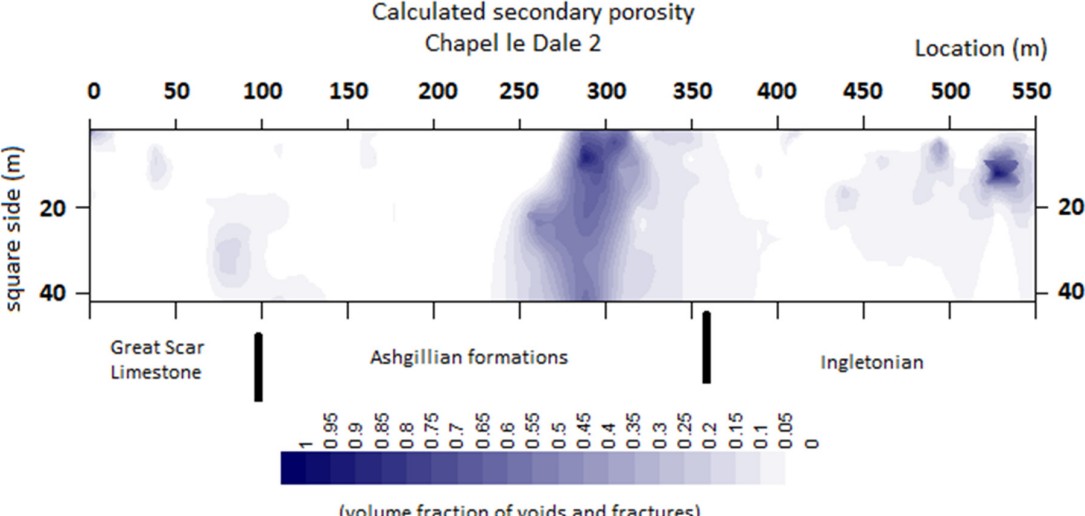

**Figure 15.** Pseudo section for the secondary porosity in CLD2. (Secondary porosity as the fraction of the total volume occupied by fractures and voids).

This pseudo section, containing all the secondary porosity values calculated for CLD2, provides more detailed information than that obtained in Figure 14.

The Great Scar Limestone boundary with the Ashgillian series is clearly shown at position 90 m. As in the discussion of the profiles in Figure 14, the contact between the Ashgillian formations and the Ingletonian is well marked, but shifted to the left, at position 300 m. The upper part of the pseudo section depicts very low secondary porosity values. This is explained as the cover is thicker in this region and, thus, the ground is behaving as homogeneous or near homogeneous nearer the surface.

Unusually high secondary porosity values are developed near the contact between the Ashgillian and the Ingletonian, clearly revealing that feature. Further higher values are shown near the surface and correspond to the close values for the calculated $\rho_{max}$ and $\rho_{min}$ or local inhomogeneities near the electrodes.

## 5. Conclusions

As only limited information is available in the literature, the results herein relate to model studies and field surveys in a region where the geology is known to investigate the drawbacks and information that secondary porosity studies can apport to the study of fracturing and the nature of formations. Such information can be very important when exploring for water in crystalline rocks and when looking for paths for fluid propagation.

The general equation for secondary porosity calculations (1) based on square array observations was adopted, and the problems concerning its use were discussed. The most important problems result from the need to know the electrical conductivity value for the groundwater in the region and the possibility of calculating very large and negative values for the secondary porosity with no geological meaning.

Unless severe aquifer contamination is present, the groundwater electrical conductivity in an area should be fairly constant. Thus, it will affect the calculated absolute secondary porosity value but, more importantly, will not affect the lateral and spatial changes of the secondary porosity, which proved to be paramount in the interpretation.

High and negative secondary porosity values were calculated in both the field and model experiments. They were checked and proven to have no geological meaning. It was found that most of them were obtained for the smallest spacings and, hence, are explained by placement errors and contact resistances from ground inhomogeneities near the electrodes. There is also the possibility of anisotropic axes rotation producing oblate anisotropy, but this effect was not found in the data, as there are no calculated strike values at $90°$ to the known geology and to the model strike. Furthermore, high values, also with no geological meaning, can also be produced in areas where $\rho_{max}$ and $\rho_{min}$ are very similar; that is, when the formations' homogeneity prevails. These data must also be checked and confirmed as they do not have geological meaning.

The computed secondary porosity proved to be very effective in the location of faults and contacts, as well as in the identification of the nature of the concealed formations. However, it must be pointed out that absolute secondary porosity values may be difficult to obtain, but, nevertheless, their lateral and spatial changes proved to be very consistent in the overall data interpretation.

Past studies have focused on the use of azimuthal resistivity techniques to determine the strike of concealed formations and fracturing. However, crossed square array measurements provide accurate estimates for the strike of concealed formations and, therefore, can be oriented in such a way that its two constituent square arrays 1 and 2, shown in Figure 1, will provide values for $\rho_{max}$ and $\rho_{min}$ to use in the secondary porosity equation.

Therefore, there is no need to use azimuthal resistivity surveys and, thus, time consuming and space demanding field surveys are not necessary. Even if the geological strike is unknown, with the aid of crossed square array soundings, it is possible to calculate that strike. Then, the survey can continue with the crossed square array oriented at $0°$ to the determined geological strike.

Future work should address models such as the contacts between isotropic blocks, faults gouges, contaminated areas, and anthropic cavities to better understand the use of secondary porosity in geology and groundwater, and to widen the applicability of these studies.

**Funding:** The author was supported by a Calouste Gulbenkian Foundation (Lisbon) grant during his stay in England.

**Institutional Review Board Statement:** Not applicable.

**Informed Consent Statement:** Not applicable.

**Data Availability Statement:** The data presented in this work can be found in the documents cited in the text with reference numbers [16,17] and in "The use of field anisotropy measurements in resistivity investigations", PhD dissertation by Manuel Matias, Univ. of Leeds, 1983.

**Acknowledgments:** The author wishes to thank Rui Moura for the assistance in the 2D modelling of the field traverses.

**Conflicts of Interest:** The author declares no conflict of interests.

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
