# Peer review of "Estimate of Secondary Porosity from Surface Crossed Square Array Resistivity Measurements"

_geosciences, doi:10.3390/geosciences13040101_

Round 1

Reviewer 1 Report

The study of secondary porosity (SP) in a terrain is the subject of this paper. Unlike invasive conventional methods, the determination of SP is carried out through the analysis of the apparent resistivity of the terrain. The crossed square array of electrodes is used as a measure of the apparent resistivity and subsequent location of geological strikes that justify the porosity. Several expressions proposed by authors from the 1970s are used for the apparent resistivity, while a confusing expression (3) is used as the basis for estimating SP.

The paper is generally well-written. It is an experimental field study that has value as a contributor of information. However, it is not clear what the contribution of this paper is to the general method employed by preceding authors. Additionally, expression (3) must be explained in much more detail, as the indicated reference [5] is inaccessible. The effective vertical anisotropy must be adequately defined in the paper in terms of the resistances measured with the square array. On the other hand, the Analog model proposed to test the methodology is confusing. Personally, I have not been able to form a mental image of it based on the description in the text and the figure (perhaps because I am not a geologist?). I would like the author to describe the model more clearly.

In summary, the paper should be revised according to the above indications before being considered for publication in Geosciences.

Author Response

Dear Reviewer,

Thank you very much for your work and suggestions. Herein I send you my comments and replies.

Reviewer 1

Thank you very much for your comments. I hope I have clarified your points. The added text and alterations are marked in green in the new revised version of the manuscript.

Below you will find my comments and answers (in italic and underlined) incorporated in your revision.

The study of secondary porosity (SP) in a terrain is the subject of this paper. Unlike invasive conventional methods, the determination of SP is carried out through the analysis of the apparent resistivity of the terrain. The crossed square array of electrodes is used as a measure of the apparent resistivity and subsequent location of geological strikes that justify the porosity. Several expressions proposed by authors from the 1970s are used for the apparent resistivity, while a confusing expression (3) this equation (in the new version equation (7) is used by previous authors from the 80’s to the 2010’s, derived from well logging. Further references have been added in line 127. The apparent resistivity and “n” equations for the square and crossed square arrays were proposed in the 70’s 80’s, have been fully used in several works from different authors and references are given. Nevertheless, some introductory theoretical concepts, equations and one new figure was added to clarify these points is used as the basis for estimating SP.

Below equation (7) the text describes its parameters and a critical review of the equation is given, including pitfalls and problems that might arise from its use, lines 129-168.

The paper is generally well-written. It is an experimental field study that has value as a contributor of information. However, it is not clear what the contribution of this paper is to the general method employed by preceding authors. Previous authors used field work surveys with no accurate knowledge of site and, even so, they did a most valuable work. To fully demonstrate the method, it is needed to apply it in areas -model and field areas- you know very well so you can address problems, issues, pitfalls, benefits and prove the method can have a broad usage or not and, in this latter case, forget about it. This is what is done here. Herein model work allows the control of field survey data and interpretation, provides material for interpretation, and the benefits and drawbacks of the technique are fully addressed. Conclusions are very critical. Pitfalls, issues are openly discussed and clues are left for future works. Furthermore, previous works used azimuthal measurements and didn’t consider the benefits of the use of square arrays and crossed square arrays. Herein it is justified the use of these arrays in order to overcome the extra field effort necessary to carry out azimuthal surveys. For the first time a thorough study of the subject is given and it is proved that the method can be used everywhere. Additionally, expression (3) must be explained in much more detail, as the indicated reference [5] is inaccessible the isbn is provided and a further reference is given in this point of the text, lines 153-154. Please look in the net for “Groundwater Geophysics Kirsch. On the 9th March it had 69k accesses and 161 citations. The effective vertical anisotropy must be adequately defined in the paper the concept is defined clearly and theoretically in terms of the resistances measured with the square array this would lead to an unnecessary extension of the paper and perhaps distract the readers from the main points of the work. Instead, the references are given. On the other hand, the Analog model proposed to test the methodology is confusing text added in the paper, dimensions given, aims and geological models given. Personally, I have not been able to form a mental image of it based on the description in the text and the figure (perhaps because I am not a geologist?). I would like the author to describe the model more clearly. Text added the features of the model are identified with equivalent geological features and the aims of the work.

In summary, the paper should be revised according to the above indications before being considered for publication in Geosciences.

Reviewer 2 Report

The manuscript presents the analogue model and real field measurements at Chapel le Dale Valley, near Ingleton, Northwest England using crossed square-array direct-current resistivity method for anisotropic underground sections. The authors use the conductivity of groundwater (microsiemens/cm) C, and  the effective vertical anisotropy “n” for graphical presentation of the numerical modelling. The manuscript has 20 pages, 26 references, 14 figures.

Comments

The title: without the point at the end. Secondary with small letters.

Instead: DC crossed square array resistivity measurements” crossed square-array direct-current resistivity method upon [12] Lane, Haeni, and Watson.

Not in the abstract, nor in the text there is no information about direct current features.

The abstract is not full. There is no mention about which models with pseudo sections? Is it for analogue model or for 2 sites at Chapel le Dale Valley, near Ingleton, Northwest England ?

L. 52 point at the end.

Please, give value units (%, Ohm·m) for all equations.

L. 96 “n” is the effective vertical anisotropy – units?

Eq. 3 - 3,14· (multiplex)

Please, keep all degrees as upper subscript not 45o but 45°. And digit-space-unit, e.g. 1.41 cm.

Line 71. Author mentions Figure 1. So please, put Figure right after this paragraph. Before the equations.

Line 76 the paradox of anisotropy [15], seems to be 16 or [15, 16].

Line 117 instead conductance, - conductivity.

Figure 2. Title near the figure. salt brine (or saline) is better. What is PVC?

Line 139. What is PVC?

All figures secondary porosity in %? So, why I see usually 0-1 secondary porosity (e.g. Figure 5), but then I see negative values (e.g. Figure 4). Why secondary porosity could have negative values? What kind of situation is this? Also why in Figures 7, 9, 11  secondary porosity is more than 1 up to 2, if it is in %, where 1 is 100 %?

I still don’t understand why array sides ranging from 1.41cm to 45.25cm with an expansion rate of √2. What is the specific step? In L. 173 is this make sense for integer and a fractional numbers 4cm, 11.31cm and 32cm? The same in Figure 9. Why fractal ends? 22.63 m (centre); 45.25 m

Figure 4. Location in cm. The is no explanation about this 60 cm length. What does it mean?

L. 204 Figure 1? ? or 2?

L. 260 θ=0°,

L. 265 Figure 6 without the point

Figures 8, 12 Ohm·m istead of Ohm.m

Figures 10, 14 units?

L. 440 Figure 1

References section . Please, check MDPI style and text font/ At this moment it is wrong, especially full names of authors.

Author Response

Dear Reviewer

Thank you very much for your comments. Below you will find my comments and answers in italic and underlined.

Reviewer 2

Thank you very much for your comments. I hope I have clarified your points. The added text and alterations are marked in yellow in the new revised version of the manuscript.

Below you will find my comments and answers (in italic and underlined) incorporated in your revision.

The manuscript presents the analogue model and real field measurements at Chapel le Dale Valley, near Ingleton, Northwest England using crossed square-array direct-current resistivity method for anisotropic underground sections. The authors use the conductivity of groundwater (microsiemens/cm) C, and  the effective vertical anisotropy “n” for graphical presentation of the numerical modelling. The manuscript has 20 pages, 26 references, 14 figures.

Comments

The title: without the point at the end. Secondary with small letters. done

Instead: DC crossed square array resistivity measurements” crossed square-array direct-current resistivity method upon [12] Lane, Haeni, and Watson. Done, title revised.

Not in the abstract, nor in the text there is no information about direct current features.

The abstract is not full. There is no mention about which models with pseudo sections? Is it for analogue model or for 2 sites at Chapel le Dale Valley, near Ingleton, Northwest England ? done

  1. 52 point at the end. done

Please, give value units (%, Ohm·m) for all equations. Done in the text.

  1. 96 “n” is the effective vertical anisotropy – units? Done, it is non dimensional

Eq. 3 - 3,14· (multiplex) done

Please, keep all degrees as upper subscript not 45o but 45°. And digit-space-unit, e.g. 1.41 cm. done

Line 71. Author mentions Figure 1. So please, put Figure right after this paragraph. Before the equations. done

Line 76 the paradox of anisotropy [15], seems to be 16 or [15, 16]. Done 15 and 16

Line 117 instead conductance, - conductivity. It is actually conductance for some authors, the ones cited

Figure 2. Title near the figure. salt brine (or saline) is better. What is PVC? done

Line 139. What is PVC? done

All figures secondary porosity in %? So, why I see usually 0-1 secondary porosity (e.g. Figure 5), but then I see negative values (e.g. Figure 4). Why secondary porosity could have negative values? What kind of situation is this? Also why in Figures 7, 9, 11  secondary porosity is more than 1 up to 2, if it is in %, where 1 is 100 %?

Ok. I understand that and this was one of the main points of the work: to understand these values. Values were checked and this issue is discussed in lines 141-152, 219 -225,281-284,330-336,405-412,481-490. As discussed, causes, other than geology, were assigned and checked to these calculations and it was added that these values have no geological meaning. It was decided to leave these values in the paper, so that, future researchers will not be deterred by them but should look for explanation and confirmation in the data. Not showing values for these spacings and locations, or substitute them by 0, for instance, even with an explanation of course is not an option as it would leave loose ends to future researchers. We never intended to avoid or forget any problem whatsoever, issues must be addressed.

I still don’t understand why array sides ranging from 1.41cm to 45.25cm with an expansion rate of √2. What is the specific step? In L. 173 is this make sense for integer and a fractional numbers 4cm, 11.31cm and 32cm? The same in Figure 9. Why fractal ends? 22.63 m (centre); 45.25 m the expansion rate is explained in the text. The fractions I understand, and I changed. However, in the field I kept the approximation to the decimetre.

Figure 4. Location in cm. The is no explanation about this 60 cm length. What does it mean? No this if Figure 5. Yes location is always in cm. I think it is shown everywherein the figure. With regard to the X axis been marked from -60 to 60 cm, it was decided to do this to keep the marks in the axis constant and to leave some space between the lines and the Y axes. This way the “n” curve doesnot intercept the YYaxis for porosity and vice versa so, in my opinion, any possible confusion is avoided to reader.

  1. 204 Figure 1? ? or 2? Done now it is figure 3
  2. 260 θ=0°, done
  3. 265 Figure 6 without the point done

Figures 8, 12 Ohm·m istead of Ohm.m done

Figures 10, 14 units? The axis have unities. Secondary porosity is a fraction of the volume so no unities, but written along the axis..

  1. 440 Figure 1 done

References section . Please, check MDPI style and text font/ At this moment it is wrong, especially full names of authors. Checked and altered.

Round 2

Reviewer 1 Report

After review by the author, the paper has improved enough to be accepted for publication in Geosciences.

Reviewer 2 Report

The manuscript looks much better. Thank you.

reference font is not Mdpi style some points at the end are missed.

keywords: crossed square array; (as a meaningful sentence)